# Solution Behavior near Envelopes of Characteristics for Certain Constitutive Equations Used in the Mechanics of Polymers

**DOI:** 10.3390/ma12172725

**Published:** 2019-08-26

**Authors:** Sergei Alexandrov, Lihui Lang, Elena Lyamina, Prashant P. Date

**Affiliations:** 1School of Mechanical Engineering and Automation, Beihang University, Beijing 100191, China; 2Ishlinsky Institute for Problems in Mechanics RAS, Moscow 119526, Russia; 3Division of Computational Mathematics and Engineering, Institute for Computational Science, Ton Duc Thang University, Ho Chi Minh City 700000, Vietnam; 4Faculty of Civil Engineering, Ton Duc Thang University, Ho Chi Minh City 700000, Vietnam; 5Department of Mechanical Engineering, Indian Institute of Technology Bombay, Mumbai 400076, India

**Keywords:** constitutive equations, interface, friction, singularity, envelope of characteristics, continuum mechanics

## Abstract

The present paper deals with plane strain deformation of incompressible polymers that obey quite a general pressure-dependent yield criterion. In general, the system of equations can be hyperbolic, parabolic, or elliptic. However, attention is concentrated on the hyperbolic regime and on the behavior of solutions near frictional interfaces, assuming that the regime of sliding occurs only if the friction surface coincides with an envelope of stress characteristics. The main reason for studying the behavior of solutions in the vicinity of envelopes of characteristics is that the solution cannot be extended beyond the envelope. This research is also motivated by available results in metal plasticity that the velocity field is singular near envelopes of characteristics (some space derivatives of velocity components approach infinity). In contrast to metal plasticity, it is shown that in the case of the material models adopted, all derivatives of velocity components are bounded but some derivatives of stress components approach infinity near the envelopes of stress characteristics. The exact asymptotic expansion of stress components is found. It is believed that this result is useful for developing numerical codes that should account for the singular behavior of the stress field.

## 1. Introduction

Qualitative behavior of solutions in the vicinity of interfaces between plastic material and rigid solids essentially depends on the constitutive equations chosen. A distinguishing feature of elastic and rigid plastic models is the existence of a yield criterion, which is a scalar constraint. Because of this constraint, some stress components are bounded. This property of the stress tensor may not be compatible with some boundary conditions. An obvious contradiction appears if the friction stress prescribed by the friction law is larger than the shear yield stress in the case of pressure-independent yield criteria. A less obvious case is associated with the regime of sticking at the interface between plastic material and rigid solids. In this case, the friction stress is determined from the solution of a boundary value problem, and its magnitude is controlled by other boundary conditions. This magnitude can attain the maximum possible value allowable by the yield criterion. Then, no solution at sticking exists and the regime of sticking should be replaced with the regime of sliding [1,2]. The conceptual difference between this regime of sliding and the regime of sliding that occurs according to a conventional friction law is that the former is fully controlled by the material model. Therefore, one can control the transition between the regimes of sticking and sliding by changing material models and parameters involved in these models.

Many models in the mechanics of polymers are based on yield functions. A generalization of the von Mises and Drucker–Prager yield criteria on polymeric materials has been proposed in [3]. The Tresca yield criterion has been modified to be applicable for polymeric materials in [4]. It has been assumed that there is a linear relationship between the first invariant of the stress tensor and the maximum shear stress. The effect of temperature on yield behavior of some polymeric materials has been studied in [5]. Strain-rate sensitivity of yield behavior of Nylon 101 has been investigated experimentally in [6]. An empirical pressure-dependent yield function has been proposed. Using molecular dynamics simulation, a multi-surface yield function has been derived in [7]. The studies above demonstrate that many polymeric materials are treated in the framework of pressure-dependent plasticity. 

When the transition between the regimes of sticking and sliding is controlled by the material model, solutions are singular at sliding. In particular, the quadratic invariant of the strain-rate tensor can approach infinity near the friction surface. This feature of solution behavior has been demonstrated in [8] for rigid perfectly-plastic material and in [9,10] for viscoplastic material with a saturation stress. Additionally, numerous analytic and semi-analytic solutions for various material models reveal this behavior of solutions [1,2,11,12,13,14,15]. It is evident that numerical solutions of the corresponding boundary value problems do not converge [16,17,18], unless a special technique is adopted (for example, [19,20]). Returning to analytic and semi-analytic solutions, approximate solutions derived by inverse methods break down under certain conditions if the assumptions concerning the velocity field are not compatible with the exact behavior of the velocity field at the interface where the transition between the regimes of sticking and sliding may or may not occur. Examples can be found in [21,22].

The discussion above shows that it is important to know the exact asymptotic behavior of the stress or/and velocity field in the vicinity of the interface between plastic material and rigid solids for both developing efficient numerical methods and solving boundary value problems approximately by inverse method. The present paper deals with yield criteria for polymers proposed/recommended in [23,24,25] and a generalization of these models under plane strain conditions. In general, the corresponding systems of equations supplemented with the equilibrium equations can be elliptic, parabolic, or hyperbolic. Henceforward, attention is concentrated on the hyperbolic regime. In this case, the solution is singular if the interface between plastic material and rigid solids coincides with an envelope of stress characteristics. A semi-analytic solution for the models proposed in [23,24] has been provided in [26]. This solution deals with the transition between sticking and sliding regimes.

The main result obtained in the present paper is valid for rate-independent models. The possibility to extend this result to rate-dependent models should depend on the way the viscosity is introduced into the model. By analogy to rate-dependent models of pressure-independent plasticity [9,10], it is reasonable to expect that singular asymptotic solutions may appear in the case of vanishing viscosity. 

## 2. Material Models

The linear and quadratic invariants of the stress tensor, $I_{1}$ and $I_{2}$, are defined as
(1)$$I_{1}=\sigma_{1}+\sigma_{2}+\sigma_{3},\;I_{2}=\sqrt{\frac{3}{2}\left(s_{1}^{2}+s_{2}^{2}+s_{3}^{2}\right)}.$$
where $\sigma_{1}$, $\sigma_{2}$, and $\sigma_{3}$ are the principal stresses and
(2)$$s_{1}=\sigma_{1}-I_{1}/3,\;s_{2}=\sigma_{2}-I_{1}/3,\;s_{3}=\sigma_{3}-I_{1}/3.$$

The yield criterion for polycarbonate is represented as [23]
(3)$$c_{1}I_{1}+I_{2}=c,$$
where $c_{1}$ and c are material constants. The yield criterion for polyvinylchloride and polycarbonate is represented as [24]
(4)$$\left(C-T\right)I_{1}+I_{2}^{2}=CT,$$
where *C* is the absolute value of the compressive yield strength and *T* is the absolute value of the tensile yield strength. Both are constant. In spite of the dependence of yielding on the linear invariant of the stress tensor, it has been shown in [23,24] that the materials tested are practically plastically incompressible. It is; therefore, reasonable to adopt the von Mises plastic potential. In this case, the flow rule in terms of the principal stresses reads
(5)$$\xi_{1}=\lambda s_{1},\;\xi_{2}=\lambda s_{2},\;\xi_{3}=\lambda s_{3},$$
where $\xi_{1}$, $\xi_{2}$ and $\xi_{3}$ are the principal strain rates and λ is a non-negative multiplier. Moreover, the principal stress and principal strain rate directions coincide. 

Equations (3) and (4) can be generalized as
(6)$$I_{2}=f\left(I_{1}\right),$$
where $f\left(I_{1}\right)$ is a prescribed function of $I_{1}$. This function is quite arbitrary. Nevertheless, it is assumed that
(7)$$df/dI_{1}<0.$$
It is evident that both (3) and (4) satisfy this inequality.

It is assumed that the material is rigid plastic (i.e., elastic strains are neglected). 

## 3. System of Equations under Plane Strain Conditions

By assumption, the flow is everywhere parallel to (x, y) planes of a Cartesian coordinate system (x, y, z) and the solution is independent of *z*. For further convenience, a curvilinear orthogonal coordinate system (t, s) is introduced in planes of flow. This coordinate system is illustrated in Figure 1, where γ+π/4 is the orientation of the principal stress $\sigma_{1}$ relative to the t− direction, measured anticlockwise positive from the t− direction. The t− curve corresponding to s=0 is given. The s− lines are straight and orthogonal to this t− curve. All other t− curves are orthogonal to the s− lines. It is always possible to introduce such a coordinate system in a vicinity of any smooth curve. It is also always possible to assume that the scale factor of the s− lines is unity. Let $\sigma_{tt}$, $\sigma_{ss}$, and $\sigma_{ts}$ be the stress components referred to the (t, s)− coordinate system. 

Then, the equilibrium equations are [27]
(8)$$\frac{\partial \sigma_{tt}}{\partial t}+H\frac{\partial \sigma_{ts}}{\partial s}+F_{1}=0,\;H\frac{\partial \sigma_{ss}}{\partial s}+\frac{\partial \sigma_{ts}}{\partial t}+F_{2}=0.$$
Here *H* is the scale factor of the t− curves and the terms $F_{1}$ and $F_{2}$ are independent of stress derivatives. Let $\xi_{tt}$, $\xi_{ss}$, and $\xi_{ts}$ be the strain rate components referred to the (t, s)− coordinate system. Then [27],
(9)$$\xi_{tt}=\frac{1}{H}\frac{\partial u_{t}}{\partial t}+u_{s}G_{1},\;\xi_{ss}=\frac{\partial u_{s}}{\partial s},\;2\xi_{ts}=\frac{1}{H}\frac{\partial u_{s}}{\partial t}+\frac{\partial u_{t}}{\partial s}+u_{t}G_{2}.$$
Here $u_{t}$ and $u_{s}$ are the velocity components referred to the (t, s)− coordinate system, and $G_{1}$ and $G_{2}$ are independent of these components and their derivatives. 

One of the principal strain rates vanishes under plane strain conditions. Assume that $\xi_{3}=0$ and that the direction of the principal stress $\sigma_{3}$ coincides with the *z*-direction of the Cartesian coordinate system. Then, it is seen from (2) and (5) that
(10)$$s_{3}=0\;\mathrm{and}\;\sigma_{3}=I_{1}/3.$$
Combining the first of these equations and the identity $s_{1}+s_{2}+s_{3}=0$ gives $s_{1}=-s_{2}$. It is possible to assume, with no loss of generality, that $s_{1}>s_{2}$. Then, $s_{1}>0$ and the second equation in (1) becomes
(11)$$I_{2}=\sqrt{3}s_{1}=-\sqrt{3}s_{2}.$$

The transformation equations for stress component in a plane result in (Figure 1)
(12)$$\sigma_{tt}=\frac{I_{1}}{3}-\frac{\left(\sigma_{1}-\sigma_{2}\right)}{2}\sin2\gamma ,\;\sigma_{ss}=\frac{I_{1}}{3}+\frac{\left(\sigma_{1}-\sigma_{2}\right)}{2}\sin2\gamma ,\;\sigma_{ts}=\frac{\left(\sigma_{1}-\sigma_{2}\right)}{2}\cos2\gamma .$$
Substituting (11) into (12) yields
(13)$$\sigma_{tt}=\sigma -\tau \sin2\gamma ,\;\sigma_{ss}=\sigma +\tau \sin2\gamma ,\;\sigma_{ts}=\tau \cos2\gamma ,$$
where $\sigma =I_{1}/3$ and $\tau =I_{2}/\sqrt{3}$. It is convenient to rewrite (6) and (7) as
(14)τ=g(σ), dg/dσ≡G(σ)<0.

The first two equations in (5) and (11) combine to give $\xi_{1}+\xi_{2}=0$ or
(15)$$\xi_{tt}+\xi_{ss}=0.$$
Since the material is incompressible and the material model is coaxial, it is evident from (12) that
(16)$$\frac{\xi_{tt}-\xi_{ss}}{2\xi_{ts}}=-\tan2\gamma .$$

## 4. Characteristics and Characteristic Relations

Substituting (13) into (8) and using (14) one gets
(17)$$\begin{array}{c} \left(1-G\sin2\gamma \right)\frac{\partial \sigma }{\partial t}+HG\cos2\gamma \frac{\partial \sigma }{\partial s}-2g\cos2\gamma \frac{\partial \gamma }{\partial t}-2Hg\sin2\gamma \frac{\partial \gamma }{\partial s}+F_{1}=0,\\ G\cos2\gamma \frac{\partial \sigma }{\partial t}+H\left(1+G\sin2\gamma \right)\frac{\partial \sigma }{\partial s}-2g\sin2\gamma \frac{\partial \gamma }{\partial t}+2Hg\cos2\gamma \frac{\partial \gamma }{\partial s}+F_{2}=0. \end{array}$$

One can choose the (t, s)− coordinate system such that the principal direction corresponding to the principal stress $\sigma_{1}$ is tangent to a t− curve at a point. Then, γ=−π/4 (Figure 1) at this point and the equations in (17) become
(18)$$\left(1+G\right)\frac{\partial \sigma }{\partial t}+2Hg\frac{\partial \gamma }{\partial s}+F_{1}=0,\;H\left(1-G\right)\frac{\partial \sigma }{\partial s}+2g\frac{\partial \gamma }{\partial t}+F_{2}=0.$$
Then, the equation for stress characteristics is
(19)$$\left|\begin{array}{cccc} 1+G & 0 & 0 & 2Hg\\ 0 & H\left(1-G\right) & 2g & 0\\ dt & ds & 0 & 0\\ 0 & 0 & dt & ds \end{array}\right|=0.$$The solution of this equation is
(20)$$\frac{ds}{Hdt}=\mp \sqrt{\frac{1-G}{1+G}}.$$
Here and in what follows, the upper and lower signs correspond to α− and β− characteristic curves, respectively (Figure 2). It is evident from (14) and (20) that the system of equations is hyperbolic if
(21)G>−1.
Henceforward, attention is concentrated on this regime. It is seen from (20) that the angle between the principal stress direction corresponding to $\sigma_{1}$ and each of the characteristic directions is

(22)$$\omega =\arctan\sqrt{\frac{1-G}{1+G}}\;\mathrm{or}\;\cos2\omega =G.$$

Since the orientation of characteristic curves has been determined, it is convenient for deriving the characteristic relations to choose the (t, s)− coordinate system, such that one of the characteristic directions is tangent to the t− curve at a point. Equation (17) is valid. However, γ=±ω−π/4 at the point in question. To derive the relation along α− lines, one should put γ=ω−π/4. Then, Equation (17) becomes
(23)$$\begin{array}{c} \left(1+G\cos2\omega \right)\frac{\partial \sigma }{\partial t}+HG\sin2\omega \frac{\partial \sigma }{\partial s}-2g\sin2\omega \frac{\partial \gamma }{\partial t}+2Hg\cos2\omega \frac{\partial \gamma }{\partial s}+F_{1}=0,\\ G\sin2\omega \frac{\partial \sigma }{\partial t}+H\left(1-G\cos2\omega \right)\frac{\partial \sigma }{\partial s}+2g\cos2\omega \frac{\partial \gamma }{\partial t}+2Hg\sin2\omega \frac{\partial \gamma }{\partial s}+F_{2}=0. \end{array}$$
Multiplying the first equation by sin2ω, the second by −cos2ω, and summing gives the characteristic relation of the form:(24)$$\sin2\omega \frac{\partial \sigma }{\partial s_{\alpha }}-2g\frac{\partial \gamma }{\partial s_{\alpha }}+\frac{\left(F_{1}\sin2\omega -F_{2}\cos2\omega \right)}{H}=0.$$

Analogously, putting γ=−ω−π/4 in (17) leads to the characteristic relation along β− lines of the form:(25)$$\sin2\omega \frac{\partial \sigma }{\partial s_{\beta }}+2g\frac{\partial \gamma }{\partial s_{\beta }}+\frac{\left(F_{1}\sin2\omega +F_{2}\cos2\omega \right)}{H}=0.$$

In Equations (24) and (25), $ds_{\alpha }$ and $ds_{\beta }$ are elements of length along the α− and β− lines, respectively.

The velocity characteristics and corresponding characteristic relations are found from (9), (15), and (16). In particular, these equations combine to give

(26)$$\frac{\partial u_{t}}{H\partial t}+\frac{\partial u_{s}}{\partial s}+u_{s}G_{1}=0,\;\frac{\partial u_{t}}{\partial s}+\frac{\partial u_{s}}{H\partial t}-2\cot2\gamma \frac{\partial u_{s}}{\partial s}+u_{t}G_{2}=0.$$

As before, one can choose the (t, s)− coordinate system, such that the principal direction corresponding to the principal stress $\sigma_{1}$ is tangent to an t− curve at a point. Then, γ=−π/4 (Figure 1) at this point and the equations in (26) become
(27)$$\frac{\partial u_{t}}{H\partial t}+\frac{\partial u_{s}}{\partial s}+u_{s}G_{1}=0,\;\frac{\partial u_{t}}{\partial s}+\frac{\partial u_{s}}{H\partial t}+u_{t}G_{2}=0.$$
Then, the equation for velocity characteristics is
(28)$$\left|\begin{array}{cccc} 1/H & 0 & 0 & 1\\ 0 & 1 & 1/H & 0\\ dt & ds & 0 & 0\\ 0 & 0 & dt & ds \end{array}\right|=0.$$
The solution of this equation is
(29)$$\frac{ds}{Hdt}=\mp 1.$$
Here and in what follows, the upper and lower signs correspond to ξ− and η− characteristic curves, respectively. It is evident from (29) that the angle between the principal stress direction corresponding to $\sigma_{1}$ and each of the characteristic directions is π/4. Equation (22) and condition (14) imply that ω>π/4. Therefore, the orientation of the characteristic lines is as shown in Figure 3. 

To derive the characteristic relations, it is convenient to choose the (t, s)− coordinate system, such that one of the characteristic directions is tangent to the t− curve at a point. Equation (26) is valid. However, γ=±π/4−π/4 at the point in question. To derive the relation along ξ− lines, one should put γ=0. Then, Equation (26) becomes
(30)$$\frac{\partial u_{t}}{H\partial t}+u_{s}G_{1}=0,\;\frac{\partial u_{s}}{\partial s}=0.$$
The first equation is the characteristic relation along the ξ− lines. This equation can be rewritten as
(31)$$\frac{\partial u_{\xi }}{\partial s_{\xi }}+u_{\eta }G_{1}=0.$$

A similar relation is valid along the η− lines. In particular,
(32)$$\frac{\partial u_{\eta }}{\partial s_{\eta }}+u_{\xi }G_{1}=0.$$

In Equations (31) and (32), $ds_{\xi }$ and $ds_{\eta }$ are elements of length along the ξ− and η− lines, respectively. Additionally, $u_{\xi }$ and $u_{\eta }$ are the components of velocity referred to the characteristic coordinate system. Of course, $G_{1}$ in (31) and (32) depend on the geometry of each characteristic curve. 

## 5. Asymptotic Behavior of Solutions near Envelopes of Characteristics

Consider the stress characteristics. Assume that the curve s=0 coincides with an α− characteristic curve or an envelope of α− characteristics. Then, γ=ω−π/4 on this line. In what follows, it is assumed that all stress and velocity components are bounded everywhere, all derivatives with respect to *s* are bounded at s=0, and the solution is represented by Laurent series with respect to *s* in the vicinity of the curve s=0. In this case, the variation of γ with *s* in the vicinity of the curve s=0 is represented as
(33)$$\gamma =\omega -\pi /4+\gamma_{0}s^{\chi }+o\left(s^{\chi }\right)$$
as s→0. In this equation, $\gamma_{0}$ is independent of *s* and χ is constant. Then,
(34)$$\cos\left[2\left(\gamma -\omega \right)\right]=2\gamma_{0}s^{\chi }+o\left(s^{\chi }\right)\;\mathrm{and}\;\sin\left[2\left(\omega -\gamma \right)\right]=1-2\gamma_{0}^{2}s^{2\chi }+o\left(s^{2\chi }\right)$$
as s→0. Multiplying the first equation in (17) by sin2ω, the second by −cos2ω, summing, and using (34) gives
(35)$$\begin{array}{c} \left[\sin2\omega -2G\gamma_{0}s^{\chi }+o\left(s^{\chi }\right)\right]\frac{\partial \sigma }{H\partial t}-\left[2G\gamma_{0}^{2}s^{2\chi }+o\left(s^{2\chi }\right)\right]\frac{\partial \sigma }{\partial s}-2g\left[1-2\gamma_{0}^{2}s^{2\chi }+o\left(s^{2\chi }\right)\right]\frac{\partial \gamma }{H\partial t}-\\ \left[4g\gamma_{0}s^{\chi }+o\left(s^{\chi }\right)\right]\frac{\partial \gamma }{\partial s}+\frac{\left(F_{1}\sin2\omega -F_{2}\cos2\omega \right)}{H}=0 \end{array}$$
as s→0. Multiplying the first equation in (17) by cos2ω, the second by sin2ω, summing, and using (34) gives
(36)$$\begin{array}{c} 2\cos2\omega \left[1-\gamma_{0}^{2}s^{2\chi }+o\left(s^{2\chi }\right)\right]\frac{\partial \sigma }{H\partial t}-\left[\sin2\omega +2G\gamma_{0}s^{\chi }+o\left(s^{\chi }\right)\right]\frac{\partial \sigma }{\partial s}-\left[4g\gamma_{0}^{}s^{\chi }+o\left(s^{\chi }\right)\right]\frac{\partial \gamma }{H\partial t}+\\ 2g\left[1-2\gamma_{0}^{2}s^{2\chi }+o\left(s^{2\chi }\right)\right]\frac{\partial \gamma }{\partial s}+\frac{\left(F_{1}\cos2\omega +F_{2}\sin2\omega \right)}{H}=0 \end{array}$$
as s→0.

If |∂σ/∂s|<∞ and |∂γ/∂s|<∞ at s=0 then (35) coincides with (24) and; therefore, the curve s=0 coincides with an α− characteristic curve. To study the behavior of solutions near envelopes of characteristics, one has to assume that
(37)|∂σ/∂s|→∞ and |∂γ/∂s|→∞
as s→0. Then, it follows from (33) that χ<1. Equation (35) contains the product $s^{\chi }\partial \gamma /\partial s$. It is seen from (33) that the order of this product is
(38)$$s^{\chi }\partial \gamma /\partial s=O\left[s^{\left(2\chi -1\right)}\right]$$
as s→0. None of the other terms involved in (35) may be of the same order, unless χ=1/2. Therefore,
(39)χ=1/2
and
(40)$$\gamma =\omega -\pi /4+\gamma_{0}\sqrt{s}+o\left(\sqrt{s}\right)$$
as s→0. It follows from this equation that the forth term in Equation (36) is of order $O\left(1/\sqrt{s}\right)$ as s→0. To cancel this term, one has to assume that
(41)$$\sigma =\sigma^{\left(0\right)}+\sigma^{\left(1\right)}\sqrt{s}+o\left(\sqrt{s}\right)$$
as s→0. Here $\sigma^{\left(0\right)}$ and $\sigma^{\left(1\right)}$ are independent of *s*.

Since |cot2γ|<∞ at γ=ω−π/4 if ω≠π/4, it is evident from (26) that the derivatives $\left|\partial u_{t}/\partial s\right|$ and $\left|\partial u_{s}/\partial s\right|$ are bounded at γ=ω−π/4. If ω=π/4 then $\left|\partial u_{t}/\partial s\right|\to \infty$ as s→0 [8].

## 6. Conclusions

The constitutive behavior of some polymers at large strains is adequately described by pressure-dependent yield criteria and pressure-independent plastic potentials. The system of equations for plane strain deformation is hyperbolic if the condition (21) is satisfied. However, in contrast to many rigid plastic models [11,28,29], the characteristic curves of stress and velocity equations do not coincide. This feature of the system of equations causes additional difficulties with the description of solution behavior near envelopes of characteristics. On the other hand, such descriptions are important for developing numerical codes because of possible singularities in stress and/or velocity fields. 

It has been shown in the present paper that the stress field near an envelope of stress characteristics is singular in the sense that the derivative of stress components with respect to the normal to the envelope approaches infinity. The exact asymptotic representation of stress solutions is given by (40) and (41). On the other hand, all derivatives of velocity components are bounded unless ω=π/4. In the latter case, the stress and velocity characteristics coincide as follows from (20), (22), and (29). This special case has been treated in [8].

In the case of pressure-independent plasticity, the singularity in solution behavior may significantly affect the temperature field near the singular surface [30]. This effect is due to the plastic work rate, which is involved in the heat conduction equation. The plastic work rate approaches infinity in the vicinity of characteristic envelopes together with the quadratic invariant of the strain-rate tensor and; therefore, results in a singular term in the heat conduction equation. In the case considered, it is difficult to expect the same result because the quadratic invariant of the strain-rate tensor is bounded everywhere. 

The main result found is useful for developing numerical codes based on, for example, the extended finite element method [31].

## Figures and Tables

**Figure 1 materials-12-02725-f001:**
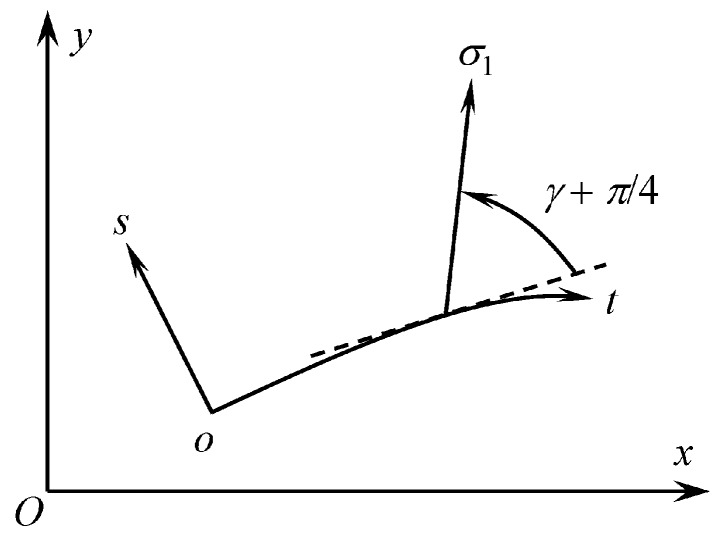
Cartesian and (*t, s*) – coordinate systems.

**Figure 2 materials-12-02725-f002:**
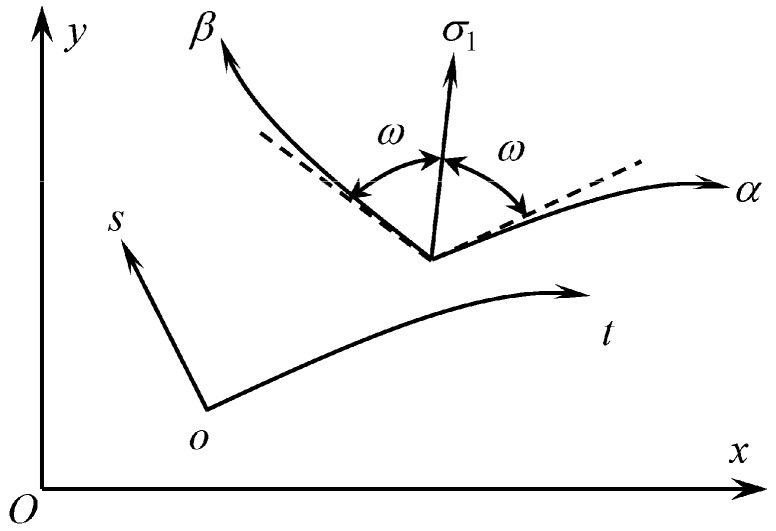
Orientation of stress characteristics.

**Figure 3 materials-12-02725-f003:**
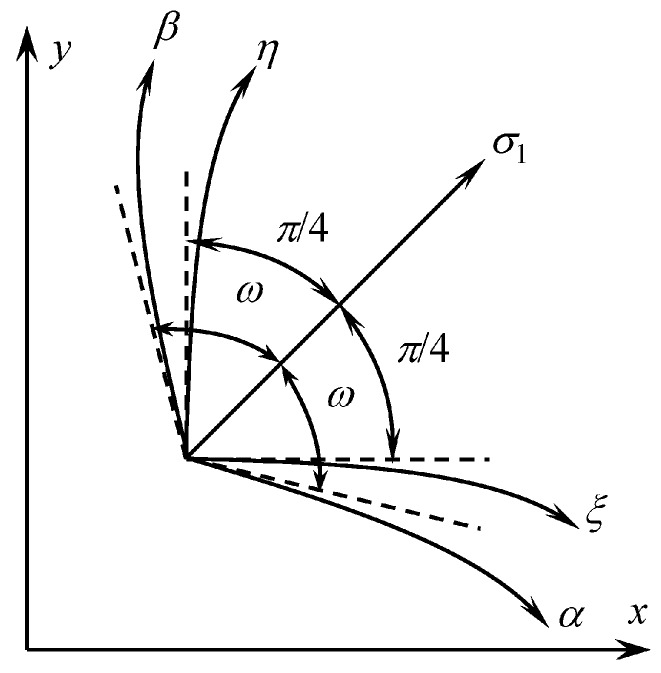
Orientation of stress and velocity characteristics.

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
