# Peer review of "Solution Behavior near Envelopes of Characteristics for Certain Constitutive Equations Used in the Mechanics of Polymers"

_materials, 2019, doi:10.3390/ma12172725_

Round 1

Reviewer 1 Report

Dear authors, 

This paper tried to provide an analytical solution to explain the mechanism of polymer deformation. However, the quality of this paper is not sufficient for publication in this journal. The presentation and description are not very clear to show the novelty of this work. Besides, with so many equations derived, it is still difficult to obtain the major contribution of this work. Most of the equations are classical in mechanics textbook. Meanwhile, no figures or plots are provided to explain the singularity issue of stress components. The hypothesis made in the manuscript is not clearly explained. In total, it is hard to read this paper and find the novelty of this work. Thus, this paper has to be rejected. 

Author Response

Please see the file attached

Reviewer 2 Report

The subject matter of the manuscript is relevant for publication in the journal and this well-written manuscript can be considered for publication after revision. I would ask that the authors to consider following recommendations for a minor revision:

1. Would the explained models be applicable for different types of polymers and their properties (for example, thermoplastic, thermoset and so on, different viscosity ranges) in solution? Please include a brief discussion in the Introduction section.

2. About one third of the only 25 citations used in this article are self-citations from the authors. Please include more references.

3. Introduce s1, s2 and s3 used in equation 1.

4. Please check the manuscript for typos and corrections. For example: “. 66.Conclusions”.

Author Response

Please see the file attached
